# Normal Maps for Rendering Vast Ocean Scenes

Category: Research

## ABSTRACT

Maritime scene simulations frequently use a height-field representation of the ocean surface. Many scenarios create a visible surface over large areas, with high amounts of detail for a camera close to the surface. Efficient rendering of a vast ocean like this makes use of Level of Detail (LOD) degradation of polygonal tessellation of the surface. But LOD degradation can have consequences in the rendered scene, particularly near the horizon, for both qualitative and quantitative metrics. The magnitude of these impacts depend on the specifics of the ocean surface conditions, and on the structure of sky light illuminating the surface. Here we present a method of extending the concept of normal mapping to efficiently restore full spatial resolution of the surface normals to the LOD degraded surface. The impact of this normal mapping process is evaluated for qualitative and quantitative metrics, across a collection of ocean surface random realizations and for a collection of sky illumination patterns. Specific cases are presented in detail, and a summary assessment of the impact of 93 simulations is presented.

**Index Terms:** Computing Methodologies—Computer Graphics—Rendering—Reflectance Modeling Computing Methodologies—Computer Graphics—Rendering—Ray Tracing Applied Computing—Physical Sciences and engineering—Earth and atmospheric sciences—Environmental sciences

## 1 INTRODUCTION

There are many graphics applications that employ realistic simulation and rendering of ocean surfaces. The VFX industry [9] and game industry [2] have applied phenomenological models of height-based ocean surfaces for a number of years. Ocean surface simulation is also used in engineering applications for ship operations trainers [3], assessment of remote sensing concepts and systems [14], and AI training of detection and classification systems [11].

The height-field approach, while present in many applications, is limited in physical fidelity because it is based on linearized Bernoulli wave theory, and so is not capable of simulating wave breaking, whitecaps, foam, or vortical motion. Its applicability to creating a vast ocean scene, meaning an ocean visible from close to camera out to the horizon, relies on phenomenological oceanographic observations of the statistical properties of ocean surfaces, treated as a random process in time and space. These empirical properties are emulated by random realizations of height fields that evolve according to a dispersion relationship, i.e. linearized Bernoulli theory of a free surface. Of course, the statistical description lacks the impact of complex nonlinear motion of the surface that only occurs transiently. Some applications supplement the height-field with a more complete CFD simulation that is either blended with the height-field surface, or driven partially by the height field realization [1, 12]. This 3D simulation is particularly useful at locations near the camera in a rendered maritime scene, but much less important at mid-range and near-horizon distances from the camera.

The creation and rendering of maritime scenes in computer graphics involves describing an ocean over regions of potentially hundreds of square kilometers. For example, for a camera located two meters above the mean ocean level, the distance to the horizon on earth is approximately 5 km, and the horizon distance for a camera at 100 meters above the mean ocean is approximately 35 km. The potentially viewable surface area for a 5 km horizon is approximately 80 km$^2$, and for a 35 km horizon, over 4000 km$^2$. Construction of a

dynamic free-surface in a 3D CFD simulation over this vast scale, with detail sufficient for a camera at a height of meters to hundreds of meters above the surface, has severe practical limitations that the height-field approximation addresses.

There are three properties of height-field ocean simulations that allow for practical construction of ocean surfaces over this vast scale. First, when the Fast Fourier Transform method [6] is used to create a patch of ocean surface height field, the properties of the FFT produce a patch of surface that is periodic and can be applied as a tile to cover any desired area, including thousands of square kilometers. Such a repetition over vast areas is known to produce visual artifacts, in which prominent waves appear in a repeating pattern. This is overcome by the second property: that multiple realizations, when created at spatially-disparate resolutions, can be added together, suppressing the repetitive artifact. This property is made possible by the fact that the height field dynamics is a linearized theory, and so several height field realizations added together is equally valid as a height field realization. By choosing the repetition rates of the individual realizations appropriately, the effective repetition distance for the combined height field can be made to be hundreds of km using only 2 or 3 realizations [7], eliminating the artifact for the distances considered in this paper. The third property relates directly to the efficiency of the task of rendering a scene with a vast ocean. When the height field is tessellated into polygons (triangles or quads), standard methods of Level of Detail (LOD) can be employed to sample the height field with larger and larger polygons as the distance from the camera increases. Tessellation allows the rendering system to use the fastest and most efficient ray intersection acceleration structures suitable for a problem. In some applications, the ray tracing task computes the intersection directly against the height field data using a type of acceleration structure [8], eliminating the need for tessellation. Ray-height-field-intersection in this fashion is not as efficient as the approach for tessellated height field. Also for height fields that have some small amount of horizontal displacement, and height fields that are "wrapped" onto curved surfaces such as a spherical earth, the performance of ray-height-field-intersection degrades, whereas ray tracing a tessellated scene is unaffected by those conditions.

The application of LOD to the pattern of tessellation has consequences, however. Larger polygons lack detail of the height variations, and so the positions of ray intersections are shifted somewhat, producing a phenomenon known as "wave hiding", i.e. there may be regions in the height field, foreground of the point of intersection with the LOD tessellated polygon, that should have intersected the ray if they had been represented by the tessellation. Ocean surface rendering typically handles light reflection and refraction using Fresnel reflectivity and transmissivity, which is very sensitive to the surface normal. But LOD tessellation loses surface normal variations across the surface of the polygon, and interpolating the vertex normals recovers very little of that detail. Because the larger polygons are distant from the camera, there can be an expectation that these losses of surface detail can have negligible impact of the rendered image. As shown in the examples of this paper, this expectation is born out in some cases, but in most circumstances there is an impact both visually and quantifiably.

This paper focuses on the issue of restoring the surface normal detail in the rendering of LOD tessellation surfaces in scenes of vast oceans, providing visual and quantitative measures of the impact of restoring that detail. The approach is to apply a variation of

the concept of normal maps [4], which are a tool for establishing detail during rendering, and for altering and controlling detail during rendering. Normal maps are typically generated and stored in a texture image. In the application here, there is no need to generate such a texture image. Instead, the original wave height simulation data can be used to generate a surface normal at any location on the surface by storing horizontal positions as vertex texture coordinates in the tessellated geometry, and reconstructing the surface normal at any location from the interpolated texture coordinate at the location of the ray-polygon intersection. No additional data is generated in preparation for rendering, and the render-time impact of on-the-fly normal construction is modest, and as noted below, can be offset in some cases by reduced time spent in construction of the ray trace acceleration structure.

To provide visual and quantitative measures of the impact of this form of normal mapping, four rendering scenarios are produced:

1. LOD tessellate with small polygons and low amounts of LOD degradation (high resolution), and render *with* normal mapping.

2. LOD tessellate with small polygons and low amounts of LOD degradation (high resolution), and render *without* normal mapping.

3. LOD tessellate with modest polygons and high amounts of LOD degradation (low resolution), and render *with* normal mapping.

4. LOD tessellate with modest polygons and high amounts of LOD degradation (low resolution), and render *without* normal mapping.

The four scenarios are created for a collection of 93 cases with randomly varing ocean surface conditions and sky illumination conditions. A visual comparison of the four scenarios for each case shows the relative impact of normal mapping and tessellation detail. Taking scenario 1 as a baseline, for each case the variance of the difference of scenario 1 and each of scenarios 2, 3, and 4 provides a quantitative assessment of the impact of normal mapping, particularly near the horizon. For this analysis, "near the horzon" is considered the range of elevations of 5 degrees below the horizon up to the horizon.

In the next section, the process of using a linear wave height field description of an ocean free surface is presented. This includes assembling the ocean from multiple "layers" of realizations, applying horizontal displacement if desired, and computing the surface normal from the combination of the surface layers. That is followed by an examination of one possible LOD tessellation process. Many tessellation schemes are possible, but the issues presented above about loss of detail in LOD tessellation apply to all, and normal mapping is applicable to all of them. In section 4 the specific implementation of normal mapping, as it applies to this specific problem, is presented, and in section 5 the impact of normal mapping on the visual and quantitative assessement is presented for a representative few of the 93 cases evaluated. The paper concludes in section 6 with an assessment of the quantitative improvements from normal mapping for all 93 cases generated.

## 2 ASSEMBLING A VAST OCEAN

Ocean surfaces represented as a height field have been in use for some time [6]. Such a representation is based on a phenomenological model of the statistical properties of the height. This leads to a Fourier-domain representation for the height field as

$$h(\mathbf{x},t) = \int \frac{d^2k}{(2\pi)^2} \, \tilde{h}(\mathbf{k},t) \, \exp{(i\mathbf{k}\cdot\mathbf{x})} \qquad (1)$$

where the height $h$ at the horizontal position $\mathbf{x} \equiv (x,z)$ on the ocean surface is the Fourier transform of a complex height amplitude $\tilde{h}$ as a function of a 2D Fourier wavevector $\mathbf{k}$. The time-dependent amplitude is assembled from random time-independent amplitudes $\tilde{h}_0(\mathbf{k})$ and a dispersion relation $\omega(k)$

$$\tilde{h}(\mathbf{k},t) = \tilde{h}_0(\mathbf{k}) \, \exp(i \, \omega(k) \, t) + \tilde{h}_0^*(-\mathbf{k}) \, \exp(-i \, \omega(k) \, t) \quad (2)$$

and $k$ is the magnitude of the 2D wavevector $\mathbf{k}$. In turn, the complex height amplitudes $\tilde{h}_0(\mathbf{k})$ are a random realization of complex values from a distribution that has a phenomenologically-prescribed spatial spectrum $P(\mathbf{k})$. There are a variety of spatial spectra that have been used for this application [9, 10].

This height field representation is sometimes supplemented with horizontal displacements of the surface, using a logic based on Gerstner waves that constructs the 2D horizontal displacement $\mathbf{D}(\mathbf{x},t)$ at any point from the height field in the Fourier space representation as

$$\mathbf{D}(\mathbf{x},t) = f_d \int \frac{d^2k}{(2\pi)^2} \left(-i\frac{\mathbf{k}}{k}\right) \tilde{h}(\mathbf{k},t) \, \exp{(i\mathbf{k}\cdot\mathbf{x})} \qquad (3)$$

and $f_d$ is a user-specified dimensionless displacement scaling parameter. With this displacement, the 3D position of the ocean surface for the "nominal" flat-plane coordinate $\mathbf{x}$ is

$$\mathbf{X}(\mathbf{x},t) = \mathbf{x} + \mathbf{D}(\mathbf{x},t) + \hat{\mathbf{y}} \, h(\mathbf{x},t) \qquad (4)$$

and $\hat{\mathbf{y}}$ is the unit vector pointing upward.

In numerical implementations, the Fourier transforms in equations 1 and 3 are replaced with Fast Fourier Transforms (FFTs), which generate the height and displacement fields on a rectangular spatial grid with user-chosen number of grid points and spatial extent, and sums over a discrete set of wave vectors that complement the number of grid points and spatial extent to the Nyquist limit. Evaluating quantities at locations that are not grid points is accomplished via bilinear interpolation. This gridded height field is also spatially periodic as a result of the FFT computation. The periodicity can be used as a tiling scheme to extend the ocean surface beyond the nominal bounds of the FFT domain. An unfortunate consequence of the periodicity of the tile pattern is that visualizations of the ocean surface can have noticeable repetitions of prominent waves in the scene. This is overcome by generating multiple random realizations of height, $h_i(\mathbf{x},t)$ and corresponding displacements, $\mathbf{D}_i(\mathbf{x},t)$ for $i = 0,\ldots,N-1$, with different choices of spatial extent and periodicity of the realizations. The full surface is assembled as the sum of these "layers":

$$h(\mathbf{x},t) = \sum_{i=0}^{N-1} h_i(\mathbf{x},t) \qquad (5)$$

$$\mathbf{D}(\mathbf{x},t) = \sum_{i=0}^{N-1} \mathbf{D}_i(\mathbf{x},t) \qquad (6)$$

When the spatial extents of the realizations are not related via integer ratios, the repetition of waves can be reduced or completely eliminated. This makes it possible to visually represent a vast ocean expanse even with only a few realizations, e.g. $N = 2$ or 3, free from repetition artifacts.

The normal for the displaced surface is computed from the expression

$$\hat{\mathbf{n}}_S(\mathbf{x},t) = \frac{\partial \mathbf{X}}{\partial x} \times \frac{\partial \mathbf{X}}{\partial z} \left/ \left| \frac{\partial \mathbf{X}}{\partial x} \times \frac{\partial \mathbf{X}}{\partial z} \right| \right. \qquad (7)$$

with partial derivatives obtained in practice either by finite differences, or by the more accurate FFT evaluation of the derivatives. For the examples shown here, the FFT approach was used to compute additional data for the spatial gradients for each layer.

Rendering a maritime scene using Global Illumination algorithms, such as Monte Carlo path tracing, is assisted by tessellating the ocean surface into polygons. Here the discussion is focused on tessellation into triangles, but the results apply equally to other choices of polygonalization. The tessellation lays out a network of grid points in the $\mathbf{x}$ coordinate, i.e. $\mathbf{x}_i$ that are arranged in collections of triangles in the flat 2D plane. Each vertex $i$ of the ocean surface tessellated geometry hold, among other possible rendering-related information, the 3D position $\mathbf{X}_i = \mathbf{X}(\mathbf{x}_i,t)$, surface normal $\hat{\mathbf{n}}_{Si} = \hat{\mathbf{n}}_S(\mathbf{x}_i,t)$, and texture coordinate $\mathbf{x}_i$. For a ray intersecting one of the triangles, the vertex normals can be interpolated to produce a surface normal at the location of the ray intersection, for use in shading and/or reflection and refraction of rays.

The choice of tessellation pattern is very dependent on the scene content within the camera field of view. For regions close to the camera, a reasonable choice is to generate triangles with sides roughly the same as the smallest grid spacing in the set of height fields used, although in some cases finer detail may be of interest because the horizontal displacements can compress together regions near the peaks of waves. However, at great distance from the camera, Level of Detail (LOD) schemes are valuable where the camera cannot resolve small triangles. In order to render a vast ocean, LOD tessellation is essential in order to efficiently render. The smallest details usually present in phenomenological ocean spectra are around a few cm in size. If triangles with 3 cm edges are tessellated from a camera located 2 m above the mean ocean surface to the horizon 5 km away, the number of triangles that must be generated is on the order of $10^{10}$, although the exact amount would depend on the choice of tessellation pattern. One LOD scheme to reduce the number of triangles is the *doubling method*, in which a user-specified "double-distance" gives the range at which the size of the triangles and the double-distance are both doubled. Figure 1 shows the doubling method for two different tessellation patterns. For example, if the triangle size close to the camera is 3 cm and the double-distance is 14 m, triangles beyond 14 m are doubled in size to 6 cm, and the double-distance is increased to 28 m. Beyond 42 m the triangles are doubled to 12 cm and the double-distance is doubled to 56 m, and so on until tessellation terminates at the furthest desired distance. In this example extending out to the horizon at 5 km, doubling happens 6 times and the triangle size at the furthest distance, near the horizon, is 1.9 m. The number of triangles is reduced from around $10^{10}$ to around $10^8$, which is a large but manageable number of triangles in current Monte Carlo path tracing software. Even so, $10^8$ triangles takes time to assemble and distribute in an acceleration structure such as a BVH tree, and has a substantial memory burden. More aggressive doubling can further reduce the resources needed. If the above example starts with triangles with resolution 3 cm and a double-distance of only 6 m, the triangle count reduces to 20% of the number for the 14 m double-distance. Similarly, keeping a double-distance of 14 m but increasing the smallest resolution to 90 cm reduces the triangle count to 3% of the original. Aggressive application of LOD has consequences, however.

## 3 CONSEQUENCES OF LOD TESSELLATION NEAR THE HORIZON

Lighting of maritime scenes can be important near the horizon. When the sun is low in the sky, the brightest part of the sky is near the horizon, and the largest gradients of the light field are near the horizon. When the sun is high in the sky, the horizon still has substantial lighting impact because of volumetric scattering of the sunlight by the atmosphere. The loss of wave detail near the horizon due to LOD tessellation could lead to biased and/or incorrect rendering of near-horizon lighting. This concern is amplified by the fact that the reflectivity, direction of refraction, and direction of reflection at ray-intersection points with water surfaces are very sensitive to the surface normal. In turn, the surface normal is ob-

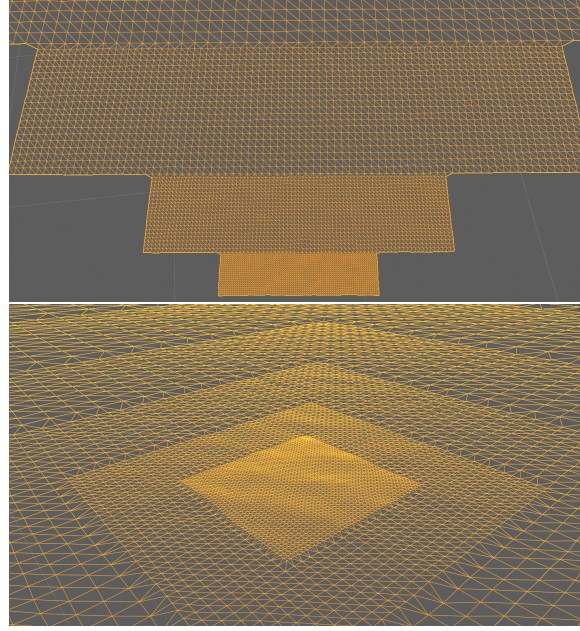

Figure 1: Two examples of tessellation patterns that use the doubling method. Top: Tessellation around the field of view of a perspective camera. Bottom: Tessellation in all directions in a square pattern of nested grids.

tained by interpolation of vertex normals, increasing the potentially negative impact of LOD tessellation. The visual impact of LOD tessellation is demonstrated in Figure 2, of an ocean surface and sky rendering in a Monte Carlo path tracer, with two different choices of the near-camera triangle size.

## 4 NORMAL MAPPING FOR OCEAN SURFACE RENDERING

A very successful way of adding and controlling detail in the shading of a surface is to inject normal maps to the shading algorithm. The normal arising from a normal map can modify or completely replace the interpolated-vertex-normal, and can be incorporate into rendering pipelines as an encoded texture. However, for rendering vast ocean surfaces with LOD tessellation, it is not necessary to generate a special-purpose texture for normal mapping, and in fact such a texture would potentially be very large when repetitive artifacts have been suppressed. Instead, we can continue to use the height $h(\mathbf{x},t)$ and displacement $\mathbf{D}(\mathbf{x},t)$ fields, composed of multiple layers $h_i$ and $\mathbf{D}_i$, and their spatial gradients. At the location of the intersection of a ray with a triangle, the texture coordinates on the triangle are interpolated to produce a nominal texture coodinate horizontal position $\mathbf{x}$ at the intersection point. This coordinate can be applied in equation 7 with the ocean realization data to compute a normal at the intersection point. This normal is used for all subsequent shading and path spawning operations.

This normal, computed on the fly at each ray-triangle intersection, contains all of the spatial detail in ocean realization. Fresnel optical properties are sensitive to the surface normal, so capturing this spatial detail has important benefits, as demonstrated in the sections below. However, it does not capture the "hiding" effects that the full height field would include, in which rays may intersect the surface earlier or later than the triangle intersection as a consequence of the lost height-field detail.

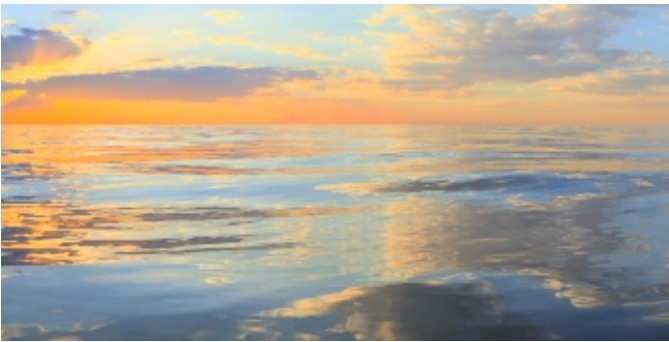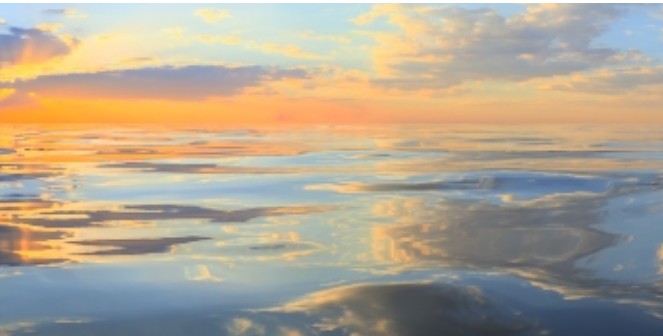

Figure 2: Two renders of the same ocean with differing amounts of tessellation. The camera is 2 m above the mean ocean surface, and tessellation extends to the horizon 5 km away. Left: Near-camera triangle size of 3 cm and double distance of 18 m (6 generations of doubling, with 1.9 m triangles near the horizon). Right: Near-camera triangle size of 90 cm and double-distance of 18 m (6 generations of doubling, with 58 m triangles at the horizon). The camera has a 360 degree field of view with equirectangular projection, and these frames are cropped from the full images.

## 5 IMPACT OF NORMAL MAPPING OCEAN SURFACE RENDERING

The examples in this paper compare two tessellations, for a variety of lighting and ocean conditions. The double-distance is 18 m for both tessellations. The high resolution tessellation has near-camera triangles 3 cm in size, which grow to 1.9 m at the horizon 5 km from the camera, for a total of 22,341,600 triangles from the camera to the horizon in all directions. The low resolution tessellation has near-camera triangles with size 90 cm, which grow to 58 m at the horizon, for a total of 25,520 triangles, roughly 0.11% of the number of triangles for the high resolution tessellation. The Monte Carlo path tracer used 1000 samples per pixel. Each intersection with an ocean surface triangle generated a Fresnel reflection and refraction. Each Monte Carlo path was limited to no more than 10 segments because initial tests found no significant additional contribution from paths with more segments. The camera has a full 360 degree spherical field of view in order to capture the impact of resolution and normal mapping throughout the environment. The only lighting of the scene was from Image Based Lighting (IBL) [5] with 360 degree sky maps composed from 360 degree photos with ground cluttered removed. Figure 3 shows two of the IBL skies used. In this images, the horizontal bisector is at the horizon, the top of the image looks straight up and the bottom of the images is straight down.

A collection of 93 variations of maritime conditions and sky IBL map were generated. The ocean realizations were based on the TMA spectrum [10], with randomly generated spectrum parameters. The IBL sky was chosen from a collection of eighteen sky maps. Here we show specific results from five of the 93 variations, chosen to illustrate the range of outcomes found. The TMA spectrum parameters for each case are in Table 1.

The variations have been evaluated based on two criteria of the impact of resolution and normal mapping near the horizon. The visual impact criterion compares the rendered images side-by-side to show the qualitative relative contributions of tessellation resolution and normal mapping. A quantitative statistical criterion treats the high resolution normal mapped render as a baseline to statistically compute the mean and standard deviation of relative luminance [13] difference between the baseline and each the other three renders, for each case. Both criteria are presented below for five chosen illustrative cases. Figures 4 and 5 show one case which demonstates the impact of normal mapping, particulary when the strongest light in the sky is near the horizon. The visual demonstration in figure 4 shows that the low-resolution tessellation with normal mapping (bottom image), produces very nearly the same image detail as the high-resolution tessellation, whether the high-resolution tessellation

case is normal mapped (middle image) or not (top image). This visual appearance is born out in the quantitative evaluation of the difference between the high tessellation, normal mapped render and each of the three other options (high tessellation without normal mapping, low tessellation with and without normal mapping). Figure 5 shows that difference, azimuthally-accumulated into mean and standard deviation of the relative luminance of the difference. The data is plotted for elevations from the horizon to five degrees below the horizon.

In this case, and in all of the cases, the green curve, representing the high tessellation without normal mapping, is most similar to the baseline high tessellation normal mapped image. Also true in all cases is that the low tessellation normal mapped result (blue curve) better matches the baseline than low tessellation without normal mapping (yellow curve). In this particular case of figure 5, normal mapping reduced the low tessellation standard deviation by a factor of 5 to 10. This case demonstrates clear and substantial impacts from normal mapping. But the impact is dependent on both ocean surface conditions and lighting conditions. The four cases below show more outcomes from variations of sky and surface parameter choices.

Figure 6 shows an environment with an overcast sky, relatively uniform intensity across the IBL image, low windspeed and low wave height. The four images (high tessellation with and without mapping, low tessellation with and without mapping) are all very similar to each other. The statistical behavior in figure 7 shows that the low tessellation cases follow each other closely, although the low-tessellation-mapped data have lower standard deviation than the low-tessellation-not-mapped data, at all elevations. Note however, that wave height is not the single key factor driving improvement from normal mapping, because the case in figures 4 and 5 is also low wave height with substantial improvement from normal mapping, while also having more variation of lighting than in the current case.

The environment in figure 8 contains a partly cloudy sky, stronger windspeed than 6, and mild wave heights. The visual improvement from normal mapping is significant. The low-tessellation-mapped result has visible differences from the baseline near the horizon, although much less pronounced that the low-tessellation-unmapped image. The standard deviation in figure 9 shows about 20% improvement from mapping.

The case in figure 11 has the same sky as figure 8 and similar windspeed and RMS wave height. But this case has much shorter fetch and much shallower bottom depth, producing ocean surface content that is smoother but with choppy waves. Visually, all four images have significant differences. The standard deviation in figure 12 is improved by normal mapping by approximately 50%, but the

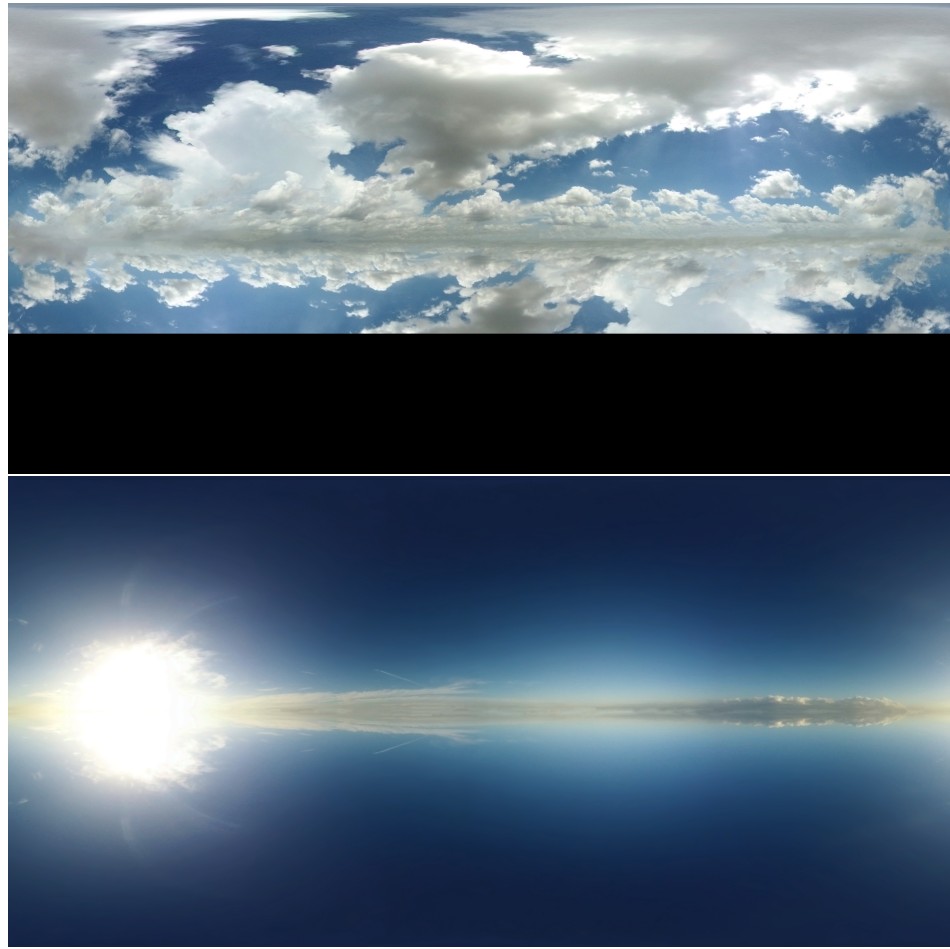

Figure 3: Two IBL skies used in rendering vast oceans. The images cover 360 degrees horizontally and 180 degrees vertically. A spherical skydome 200 km above the ocean surface uses the skymap as a texture.

Table 1: Maritime conditions for the example cases based on TMA spectrum.

| Case | Windspeed (m/s) | RMS Height (m) | Fetch (km) | Depth (m) |
|------|-----------------|----------------|------------|-----------|
| 12 | 1.3 | 0.098 | 194.8 | 381.4 |
| 22 | 4.16 | 0.225 | 204.5 | 997.9 |
| 34 | 2.38 | 0.239 | 117 | 150.9 |
| 50 | 4.91 | 0.548 | 99.95 | 678.8 |
| 87 | 1.78 | 0.127 | 141.35 | 263.4 |

Figure 4: Case 87. Renders using normal mapping of the ocean surface. Top: Near-camera triangle size of 3 cm and double distance of 18 m (6 generations of doubling, with 1.9 m triangles near the horizon) and **no normal mapping, for reference**. Top-Middle: Near-camera triangle size of 3 cm and double distance of 18 m (6 generations of doubling, with 1.9 m triangles near the horizon) with normal mapping. Bottom-Middle: Near-camera triangle size of 90 cm and double-distance of 18 m (6 generations of doubling, with 58 m triangles at the horizon) with normal mapping. Bottom: Near-camera triangle size of 90 cm and double-distance of 18 m (6 generations of doubling, with 58 m triangles at the horizon) without normal mapping.

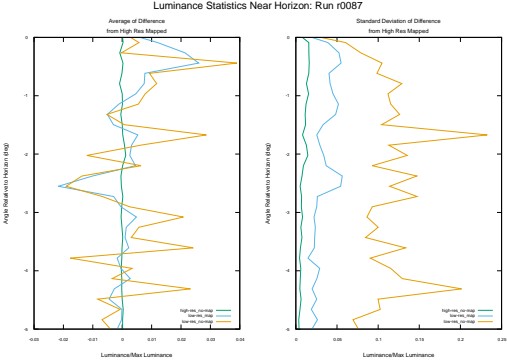

Figure 5: Case 87. Left: Azimuth-averaged mean difference from the high resolution mapped case. Right: Azimuth-averaged standard deviations from the high resolution mapped case. Green: High resolution unmapped; Blue: low resolution mapped; Yellow: low resolution unmapped.

Figure 6: Case 12. From top to bottom: High resolution with normal map, high resolution without normap map, low resolution with normal map, low resolution without normal map.

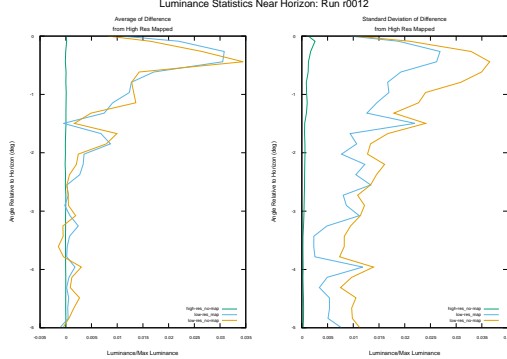

Figure 7: Case 12. Left: Azimuth-averaged mean difference from the high resolution mapped case. Right: Azimuth-averaged standard deviations from the high resolution mapped case. Green: High resolution unmapped; Blue: low resolution mapped; Yellow: low resolution unmapped.

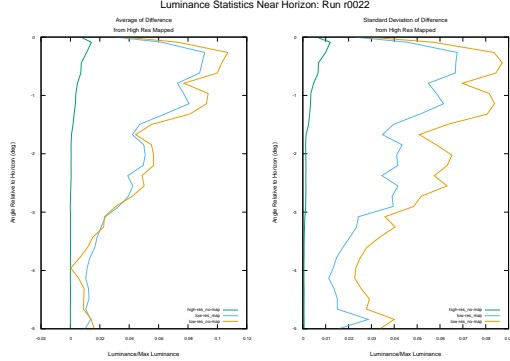

Figure 8: Case 22. From top to bottom: High resolution with normal map, high resolution without normap map, low resolution with normal map, low resolution without normal map.

Figure 9: Case 22. Left: Azimuth-averaged mean difference from the high resolution mapped case. Right: Azimuth-averaged standard deviations from the high resolution mapped case. Green: High resolution unmapped; Blue: low resolution mapped; Yellow: low resolution unmapped.

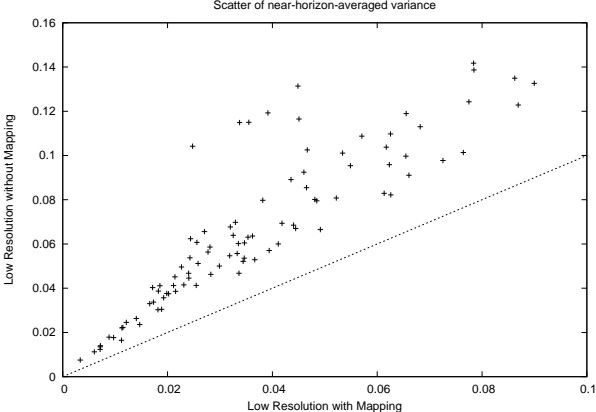

Figure 10: Scatter plot of azimuth-averaged standard deviation, averaged over angles from 0 to 5 degrees below the horizon, for 93 cases. The six outlier points at the top of the scatter are cases with the same sky as 87 (figure 4) but varying ocean surface conditions. The straight line is equal variance with and without mapping. Points below the line have lower standard deviation when not mapped, and points above the line have lower standard deviation when mapped.

low-tessellation outcomes are statistically very different from the high-tessellation outcomes. This case is the kind of scenario that might be impacted by the lack of wave hiding in the near-horizon region with heavy loss of surface detail from LOD degradation.

Another case that may suffer from insufficient wave hiding is shown in figure 13. This case has extensive clouds while not being overcast, the highest windspeed, and more than twice the RMS wave height as the other cases. While the fetch is relatively short, the depth is large and the waves are not smooth like that of figure 11. The normal mapping improves the standard deviation in figure 14 by a factor of 2 near the horizon. This case also illustrates that normal mapping a low-tessellation case can also improve the visual result near camera (toward the bottom of the images).

## 6 CONCLUSIONS

The impact of normal mapping is summarized in the scatter plot in figure 10. This plot compares the azimuthally-accumulated standard deviation of the luminace difference, averaged over the near-horizon ( 0 to -5 degrees) of the low resolution tesselled renders with and without mapping, for all 93 cases generated. One feature is the six "outlier" cases with much better performance, lying above the scatter. Figure 4 is one of them, and all six have the same sky but varying ocean conditions.

In all 93 cases, normal mapping has brought renders with low-resolution tessellation closer statistically to the high-resolution tessellation. The high standard deviation cases have the highest wave heights among the cases studied. As noted for cases 34 and 50, these high-standard-deviation cases may suffer from lack of wave hiding on large triangles near the horizon as one source for their higher values.

The resource requirements of this approach to normal mapping are modest, and in some cases favorable. The calculation of normals on-the-fly at each ray-triangle intersection added between 1% and 11% to the total render time for the various cases. However, the low-resolution tessellation cases saved an amount of time building the acceleration tree structure, in this study a BVH tree, that was

comparable to, and sometimes more than, the additional time-cost of normal calculations.

In every one of the 93 cases studied, normal mapping improved the quality of the rendered result, both visually and statistically. The extent of the improvement was sensitive to the sky light and the ocean surface conditions. The application of this approach to normal mapping for ocean surface renders may have a systematic benefit under routine use.

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

Figure 11: Case 34. From top to bottom: High resolution with normal map, high resolution without normap map, low resolution with normal map, low resolution without normal map.

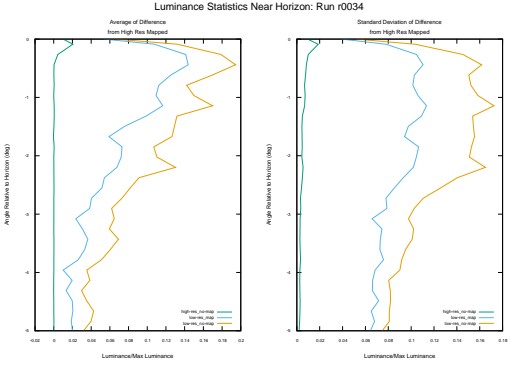

Figure 12: Case 34. Left: Azimuth-averaged mean difference from the high resolution mapped case. Right: Azimuth-averaged standard deviations from the high resolution mapped case. Green: High resolution unmapped; Blue: low resolution mapped; Yellow: low resolution unmapped.

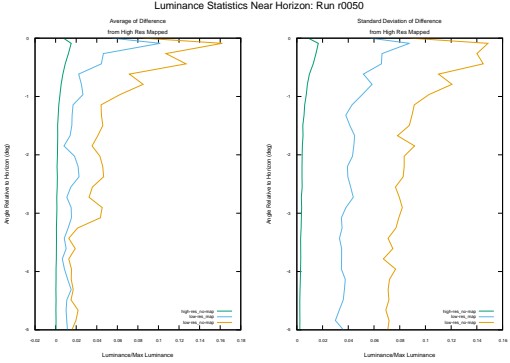

Figure 13: Case 50. From top to bottom: High resolution with normal map, high resolution without normap map, low resolution with normal map, low resolution without normal map. In this case, the near-camera structue was also improved by normal mapping.

Figure 14: Case 50. Left: Azimuth-averaged mean difference from the high resolution mapped case. Right: Azimuth-averaged standard deviations from the high resolution mapped case. Green: High resolution unmapped; Blue: low resolution mapped; Yellow: low resolution unmapped.

