# OpenReview forum: "Normal Maps for Rendering Vast Ocean Scenes"
_graphicsinterface.org/Graphics_Interface/2022/Conference — Submitted to GI 2022_

### Official Review · Reviewer_d3vY · 2021-12-30
**Interesting study, limited contribution**

**Rating:** 4
**Confidence:** 4

**Review:**

The paper presents a quantitative comparison for the rendering of oceans based on the tessellation of height fields. The ocean height is computed based on a standard mixture of Fourier functions. The main result from the analysis is that low-resolution tessellation with the use of a normal map sometimes compares favorably to the use of a reference of a high-resolution tessellation.

The paper does not compare enough to related work on tessellation and ocean rendering. Good starting points include:
* Real-time Realistic Ocean Lighting using Seamless Transitions from Geometry to BRDF, Bruneton Éric, Neyret Fabrice, Holzschuch Nicolas, Comput. Graph. Forum, 29 (2), 487-496, 2010.
* Szirmay-Kalos, L. and Umenhoffer, T. (2008), Displacement Mapping on the GPU — State of the Art. Computer Graphics Forum, 27: 1567-1592.
* Frank Losasso and Hugues Hoppe. 2004. Geometry clipmaps: terrain rendering using nested regular grids. In ACM SIGGRAPH 2004 Papers (SIGGRAPH '04). Association for Computing Machinery, New York, NY, USA, 769–776.
* Michael Doggett and Johannes Hirche. 2000. Adaptive view dependent tessellation of displacement maps. In Proceedings of the ACM SIGGRAPH/EUROGRAPHICS workshop on Graphics hardware (HWWS '00). Association for Computing Machinery, New York, NY, USA, 59–66.
* Darles, E., Crespin, B., Ghazanfarpour, D. and Gonzato, J. (2011), A Survey of Ocean Simulation and Rendering Techniques in Computer Graphics. Computer Graphics Forum, 30: 43-60.
* Hu, Y., Velho, L., Tong, X., Guo, B. and Shum, H. (2006), Realistic, real-time rendering of ocean waves. Comp. Anim. Virtual Worlds, 17: 59-67.
* Xudong Yang, Xuexian Pi, Liang Zeng and Sikun Li, "GPU-based real-time simulation and rendering of unbounded ocean surface," Ninth International Conference on Computer Aided Design and Computer Graphics (CAD-CG'05), 2005, pp. 6

I feel it is hard to grasp what is the contribution of the paper. To me, it feels it is “only” the comparison.

The paper does not have a “Related Work” section. This also makes it hard to understand the contribution against state-of-the-art papers.

A comparison to Proland would also be quite relevant/important: https://proland.inrialpes.fr/index.html

I feel that the presented study could be useful, but I also feel it is the only significant contribution, so it might be thin for a conference paper at GI. Would be very interesting as a poster though.

I feel it is hard to get clear conclusions about when the low resolution with normal mapping is good enough and when it is not. Probably a grid or matrix-like table showing sky type against wave types, with performance of low resolution with normal mapping would be easier to grasp than the current textual explanation.

Other than the above, the paper is well written and easy to understand.

More stats comparing rendering time for low resolution with normal mapping vs. high resolution would be useful.

I feel the paper should state right from the beginning that the intent is offline rendering and neglects animations/temporal artifacts.

I wonder why deviation of luminance was used instead of other metrics such as PSNR. I would also think that SSIM would be even better as it is perceptual.

Fig. 3 takes lots of space, but adds little value. I would significantly reduce its size.

Fig. 4, 6, 8, 11, and 13, while interesting, take a lot of space. I would select a subset.

Conversely, Fig. 5, 7, 9, 12, and 14 show the important/interesting results, but the fonts are very small (need to zoom to read). While enlarging the graphs is probably not necessary, enlarging the fonts is. I suggest moving the top title to the caption and arranging the secondary titles on a single line. It would be possible to fit one such graph in a single column (or a pair of such graphs on two columns).

I would add a figure like Fig. 10, but plotting low resolution against the reference for the 93 cases.

I was surprised by the lack of resolution of the renderings. For example, just zooming a little on Fig. 2 reveals big pixels.

Minor comments:
* Typo: “In this images”
* RMS is not defined
* Extraneous space in “( 0 to -5 degrees)”

---

### Official Review · Reviewer_cDrL · 2022-01-14
**Review for Normal Maps for Rendering Vast Ocean Scenes**

**Rating:** 7
**Confidence:** 3

**Review:**

The paper presents an adapted LOD technique for rendering oceans represented by height fields. The main insight for the paper is that the conventional approximation (i.e., LOD) can introduce a degraded rendering result for the specific application. In the application, surface normals have high-frequency details, and simply averaging the normals by a conventional LOD scheme leads to too blurry reflections. It motivates the authors to design a technique that restores the surface normals for an improved LOD rendering of oceans.

The paper presents a simple idea for estimating details of surface normals at a location (i.e., a ray-polygon intersected point), which uses the original simulation data. This approach is similar to the classical normal mapping, where a normal is fetched from a textured image whenever it is required. Nevertheless, introducing the normal mapping to the specific application is interesting.

The normal mapping idea is evaluated with extensive data, and it indicates that introducing the normal mapping idea to the LOD rendering of ocean scenes can amend the lost details (e.g., high-frequency reflections on the water) when exploiting low-resolution height fields.

Overall, I think the idea is not entirely novel as it merges the two well-known rendering techniques (LOD and normal mapping), but the motivation and specific use case for rendering heightfields can be considered a good contribution.

As minor comments, I would suggest the authors refine the writings and figures so that this paper can be more readable. For example, the figure legends in the plots (e.g., Fig. 5) are too small. Also, I do not think using two-column figures for plots (e.g., Fig. 5 and 7) is necessary.

---

### Official Review · Reviewer_TmcG · 2022-01-15
**Paper requires major revisions to elucidate contributions and results**

**Rating:** 3
**Confidence:** 2

**Review:**

I do not believe that this paper is ready for publication. There are a few main points that I think the authors could address to improve their paper for resubmission.

1. Clearly state the goal of the paper

After reading the paper, I'm still not really sure what problem the paper is trying to solve and what the results are. There is a lot of space dedicated to the subtleties of camera angles, triangle numbers, doubling distance configurations, etc. I would encourage the authors to move this into one, self contained subsection in a results section. The point of the paper should made crystal clear in the introduction. The paper focuses on surface normal details for large ocean scene, then I would encourage the authors to make that point in the first or second paragraph of the introduction and focus on the LOD part.

2. Compare the papers contributions to previous work

There doesn't seem to be a comprehensive related work section. I would expect to see comparisons to the works by Tessendorf. Since FFT-based waves and normal mapping are fairly established techniques, I would like to understand how the work presented in this paper is different. If the paper is purely a new method for using LOD, that contribution should be front and center. As it stands, I don't quite know what the contribution is.

3. Results need context

There are several rendered images that all generally look the same. Please help the reader here and explicitly say what is going on. State what image is the "ground truth", what images are using approaches that lead to artifacts, what approaches are too slow, etc. The captions don't provide much of a guide to tease out the results. (I would also suggest naming the various triangle size, double distance, etc. examples with some sort of shorthand. It's cumbersome to read all these numbers and try to remember the context).

4. Summarize the results

The results are provided without a clear narrative. What would you like the reader to take away from this paper? How does table 1 contribute to my understanding of paper's method and the value of it? Similar for the stats plots. What am I supposed to be looking for? If the point of the paper was stated clearly in the introduction, this might be easier to interpret but I would still encourage the authors to layout very clearly what these plots are showing and what they are seeing (is the mapped and unmpapped plots supposed to be close together? Is that good or bad?)

5. Conclusion should be a summary

When reading the conclusion, I was hoping to find the following -- the problem you solved, the method you used, the contributions of your paper, and the benefit of your new method (summary of the results). The conclusion as it is written is confusing. After reading it, I still don't know what the goal of this paper is.


In conclusion, I believe this paper requires major revisions for a resubmission. I was not able to adequately evaluate the contributions of this paper in its current form.

---

### Decision · Program_Chairs · 2022-01-18

Reject